# MCF-Net: Fusion Network of Facial and Scene Features for Expression Recognition in the Wild

**Hui Xu [1]**, **Jun Kong [1,2]**, **Xiangqin Kong [1]**, **Juan Li [2,\*]** and **Jianzhong Wang [1,3,\*]**

1 College of Information Science and Technology, Northeast Normal University, Changchun 130117, China
2 School of Social Welfare, Changchun Humanities and Sciences College, Changchun 130117, China
3 Key Laboratory of Applied Statistics of MOE, Northeast Normal University, Changchun 130024, China
\* Correspondence: seesawyy@126.com (J.L.); wangjz019@nenu.edu.cn (J.W.);
   Tel.: +86-19975860625 (J.L.); +86-18686499470 (J.W.)

**Abstract:** Nowadays, the facial expression recognition (FER) task has transitioned from a laboratory-controlled scenario to in-the-wild conditions. However, recognizing facial expressions in the wild is challenging due to factors such as variant backgrounds, low-quality facial images, and the subjectiveness of annotators. Therefore, deep neural networks have increasingly been leveraged to learn discriminative representations for FER. In this work, we propose the Multi-cues Fusion Net (MCF-Net), a novel deep learning model with a two-stream structure for FER. Our model first proposes a two-stream coding network to extract face and scene representations. Then, an adaptive fusion module is employed to fuse the two different representations for final recognition. In the face coding stream, a Sparse Mask Attention Learning (SMAL) module is utilized to adaptively generate the corresponding sparse face mask according to the input image. Meanwhile, we employ a Multi-scale Attention (MSA) module for extracting fine-grained feature subsets, which can obtain richer multi-scale interaction information. In the scene coding stream, a Relational Attention (RA) module is applied to construct the hidden relationship between the face and contextual features of non-facial regions by capturing the pairwise similarity. In order to verify the effectiveness and accuracy of our model, a large number of experiments are carried out on two public large-scale static facial expression image datasets, CAER-S and NCAER-S. By comparing the performance of our MCF-Net with other methods, the proposed model achieves superior results on two in-the-wild FER benchmarks: CAER-S with an accuracy of 81.82% and NCAER-S with an accuracy of 45.59%.

**Keywords:** expression recognition; sparse mask attention learning; multi-scale attention; Relation Attention; feature fusion

## 1. Introduction

Normal communication between humans involves both verbal and non-verbal communication. As one of the most powerful, natural, and universal non-verbal ways for human beings to convey their emotional states and intentions [1,2], human facial expressions are extremely essential in social communication. As a result, automatic facial expression recognition is important for many human–computer interaction applications, and various computer vision-based studies have been proposed to deal with this problem. However, many facial expression recognition (FER) methods in the early studies were trained and tested with a small amount of data in a controlled environment. Thus, although they achieved good results on some specific datasets, the generalization performance of these methods is not ideal. Recently, the focus of FER has shifted from laboratory-controlled scenarios to in-the-wild conditions after some related datasets have been collected. Nevertheless, the complex factors in the wild such as occlusion, posture variance, low-quality facial images, and the interference of the background will deteriorate the recognition accuracy, which brings great challenges to FER.

In traditional FER methods, facial expression features are usually captured by some manually designed feature extractors [3,4], and then classified by classical techniques [5,6]. Ref. [7] proposed an efficient human expression recognition method in the transformed domain using Discrete Contourlet Transform (DCT). Ref. [8] introduced an approach that combined specialized pairwise classifiers trained with different feature subsets for facial expression classification. Ref. [9] focused on extracting features from fused holistic and geometrical facial expression feature vectors. Ref. [10] presented an approach for facial expression recognition from images using the principle of sparse representation with a learned dictionary. Ref. [11] put forward an efficient and fast facial expression recognition system with a well-designed tree-based multi-class Support Vector Machine (SVM) classifier. In traditional methods, the dimensionality of extracted facial expression features is high. Therefore, various dimensionality reduction algorithms are usually utilized for refining the features. However, the shallow dimensionality reduction may lead to poor robustness and deteriorate the generalization ability of the methods.

Compared with traditional methods, the deep network can be trained in an end-to-end manner, so that feature learning and classification can be accomplished simultaneously. Thus, some deep learning-based FER methods have been proposed and achieved superior performance compared to traditional methods. Among these methods, the deep Convolution Neural Network (CNN) [12–17] is the most used technique for feature extraction of facial expression. Ref. [18] combined a Decision Tree (DT), Multi-Layer Perceptron (MLP), and CNN to recognize facial expressions using different imbalanced datasets. Ref. [19] employed a deep CNN to construct a facial expression recognition system, which was capable of discovering deeper feature representations of facial expressions to achieve automatic recognition. Ref. [20] represented an Identity-aware Convolutional Neural Network (IACNN) to alleviate variations introduced by personal attributes. Ref. [21] proposed a human-centric CNN architecture by using regional images for emotion recognition in static images. The aforementioned works showed that replacing the hand-crafted feature extractors with a deep CNN can effectively improve the performance of FER. Thus, FER methods based on deep learning are an important development trend. However, most research on facial expression recognition is conducted on relatively small datasets acquired in a highly controlled environment. Thus, although they work well in these well-controlled databases, their performances on real-world expression recognition tasks are usually unsatisfactory.

In order to extract more fine-grained facial expression features, current methods generally use manual and mechanical segmentation to crop the face image into a number of regions for extracting local features of facial regions [22,23]. These methods need a lot of experiments to determine the number of blocks, which are inefficient and lead to a large increase in the number of parameters. Furthermore, typical convolutional networks apply all convolutional filters on the entire image. However, the expression we want to classify or detect is surrounded by background pixels, and the necessary features can be extracted using only a few operations. This spatially sparse characteristic in the images will also impair the effectiveness of existing methods.

To deal with the above problems, we propose a novel deep learning model with the two-stream structure named Multi-cues Fusion Net for FER. Inspired by the fact that both facial and scene information is important for FER in the wild [24], our model constructs two networks for the extraction of multi-cue features. In the face coding stream, the facial images are first convoluted and max-pooled. To further study the effect of different encoding network architectures, ResNet-18 [15] is adopted to replace the shallow convolution and pooling layers of the face coding stream. Then, the Sparse Mask Attention Learning (SMAL) module [25] is introduced to extract facial expression features. SMAL can adaptively learn the corresponding key mask region, which defines the vital spatial positions to be processed by convolutions. Due to the fact that the occlusion and pose variation problems lead to a significant change in facial appearance at the spatial level, the Multi-scale Attention module [26] is applied to extract features with different receptive fields for increasing the robustness of our method to deal with the diversity of global

features. In addition, channel attention [27] is introduced to make the network pay attention to more discriminative and significant features under the channel dimension. The face coding network can extract more comprehensive and key features of facial expression and eliminate the interference of useless information, so as to enhance the feature extraction ability of the network. Although the facial region is the most informative representation of human emotion, scene information also plays an important role in the understanding of the perceived emotion, especially when the emotions on the face are expressed weakly or indistinguishable. Consequently, in the scene coding stream, the scene images without the facial region are input into the context coding module for collecting contextual cues other than facial expressions. Here, the Relational Attention module [28] is utilized to capture the similarity between the facial features and contextual features of non-facial regions to model the hidden relationship. Finally, the facial expression features and context features are combined by the adaptive fusion network for classification.

The contributions of this paper can be summarized as follows:

1. A novel deep learning model with a two-stream structure, named MCF-Net, is proposed for facial expression recognition.
2. In the face coding stream, the Sparse Mask Attention Learning module can adaptively generate the key sparse mask, which avoids manual and mechanical segmentation to extract facial features. Furthermore, the Multi-scale Attention module is employed to extract multi-scale global features at a fine-grained level and richer channel information.
3. In the scene coding stream, the Relational Attention module captures more important contextual information about non-facial regions and establishes the connection between the facial and contextual features, so as to guide the network to focus on more meaningful regions in the real-world scene.
4. A large number of experiments are carried out on two standard FER datasets to evaluate the effectiveness and feasibility of our proposed model. By comparing the performance of our MCF-Net with some existing FER methods, the advantage of the proposed method can be demonstrated.

## 2. Related Work

In the field of computer vision and machine learning, various facial expression recognition systems have been explored to encode expression information from facial representations. The complexity of head posture, occlusion, and illumination will affect the results of FER. Therefore, some scholars are committed to utilizing deep learning methods for FER. FER methods based on deep learning can be divided into two main categories according to the image acquisition forms: FER methods in the controlled environment and FER methods in the wild.

### 2.1. FER Methods in the Controlled Environment

Many researchers have created many effective ways to improve the FER results in a controlled environment, while largely relying on well-defined databases. Ref. [29] proposed a deep learning approach based on an attentional convolutional network to focus on important parts of the face and achieved improvement on FER. Ref. [30] presented an end-to-end network architecture with a Gaussian space representation for expression recognition. Ref. [31] introduced a simple yet efficient Self-Cure Network (SCN) that suppressed uncertainties efficiently and prevented deep networks from over-fitting. Ref. [32] used a landmark-guided attention branch to find and discard corrupted features from occluded regions so that they were not used for recognition. Ref. [33] developed a convolution neural network with an attention mechanism that could perceive the occlusion regions of the face and focus on the most discriminative un-occluded regions. Ref. [34] proposed an occluded expression recognition model based on the generated countermeasure network with two modules, namely occluded face image restoration and face recognition. Some other recent works on facial expression recognition included multiple networks for facial expression

recognition [35,36] and a deep self-attention network for facial emotion recognition [37]. Although these works performed reasonably well in a controlled condition, their performance may be influenced by many factors, such as illumination, partial faces, and image variation in a real scene. Therefore, the methods designed for the controlled environment will fail to perform well on more challenging datasets. Moreover, the small size of datasets constructed in controlled conditions for expression recognition research will also make the training of deep networks very challenging.

### 2.2. FER Methods in the Wild

Due to various restrictions and problems in the controlled environment, some deep learning-based works have been presented for facial expression recognition in the wild. Besides, techniques such as pretraining, multi-modal data, and GAN have also been applied to solve the problem of overfitting in the training process. Ref. [38] presented FaceNet2ExpNet to train an expression recognition network based on static images. Ref. [39] proposed recognizing video emotions in an end-to-end manner based on a deep Visual-Audio Attention Network. Ref. [40] introduced a learning-based algorithm for context-aware perceived human emotion recognition by combining three interpretations of context. Ref. [41] designed an Expression Generative Adversarial Network (ExprGAN) for photo-realistic facial expression editing with controllable expression intensity. Ref. [42] proposed a feature separation model exchange-GAN for the FER task, which can realize the separation of expression-related features and expression-independent features. Although fine-tuning and GAN can partially alleviate the issue of small datasets, the performance is still relatively low as the deep features likely contain redundant information.

More recently, some FER models based on CNN leveraged auxiliary modules or the combination of multiple networks to further enhance the feature extraction ability of the network. Ref. [43] represented a fusion framework for static facial expression recognition in the wild by varying multiple network architectures, input normalization, weight initialization, and several learning strategies. Ref. [44] introduced a Meta Auxiliary Learning method (MAL) that automatically selected highly related facial expression samples by learning adaptive weights for the training facial expression samples in a meta learning manner. Ref. [45] designed a local-feature extractor and a channel-spatial modulator to improve the robustness of a lightweight network for FER, in which the depthwise convolution was employed for local and global-salient facial features. Ref. [46] used a transformer-based cross-fusion paradigm that enabled effective collaboration of facial landmarks and direct image features to maximize proper attention to salient facial regions. Ref. [47] adopted Facial Landmark Detection (FLD) as the auxiliary task and explored new multi-task learning strategies for FER. Ref. [48] proposed an Adaptive Correlation Loss to guide the network towards generating embedded feature vectors with high correlation for within-class samples and less correlation for between-class samples. The above methods usually improve the performance of the models by increasing the network width or deepening the network depth, which leads to a large number of parameters and high computational complexity. Furthermore, the above methods only take the facial regions into consideration, which may weaken their performances once the human facial regions in images are partly occluded or blurred. Unfortunately, occlusion caused by pose variations or other objects and blur induced by human movement are common problems for the FER task in the wild. Hence, in order to extract more comprehensive and compensatory features, various kinds of cues, such as hands, body posture, and interaction with others, should be exploited.

### 3. Materials and Methods

FER has received increasing interest in the computer vision community. Although existing facial expression classifiers achieve satisfying results in analyzing constrained frontal faces, they fail to perform well in the wild. FER in the wild is challenging due to various unconstrained conditions. The architecture of our method is shown in Figure 1. The proposed model takes static images of large-scale natural scenes as input. To take both facial

expressions and contextual cues into account, the backbone of our MCF-Net is divided into two streams. The two-stream network architecture is composed of a face coding stream and a scene coding stream, which are used to extract facial expression information and context information, respectively.

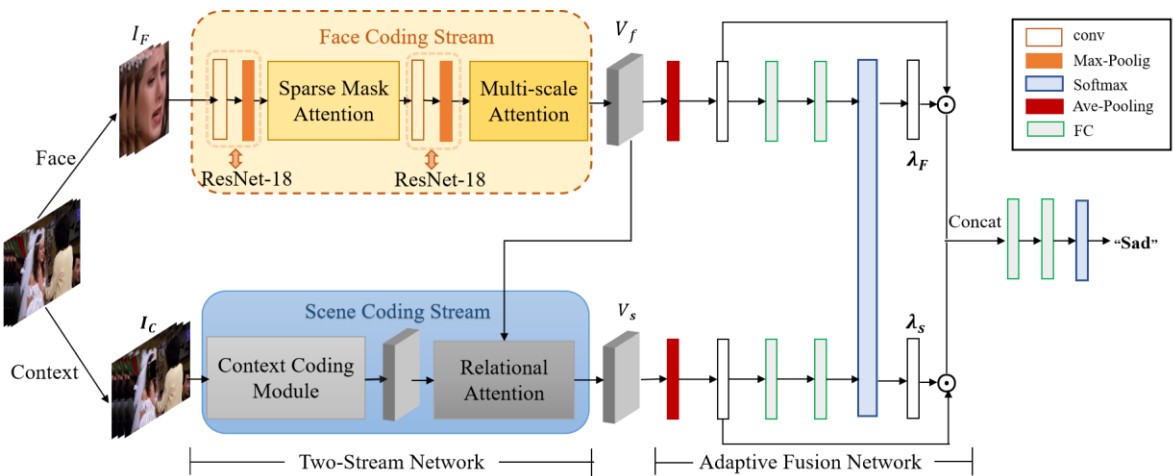

**Figure 1.** The architecture of our proposed method.

The images of facial expressions detected by the Dlib algorithm [49] are used as the input of the face coding stream. The face coding stream consists of $3 \times 3$ convolutions with batch Normalization (BN) and the ReLU activation function, max-pooling with the stride of 2, the SMAL module, and the MSA module. Here, we also attempt to substitute the shallow convolution and pooling layers of the face coding stream with ResNet-18. The input of the scene coding stream is the images without the facial regions. The scene coding stream includes the context coding module and the relational attention module. The context coding module is composed of a 5-layer $3 \times 3$ convolution with the BN and ReLU activation function, and 4-layer max-pooling with the stride of 2. In the adaptive fusion network, the full-connection and softmax layers are utilized to fuse facial expression features and contextual features by automatically learning their corresponding weights for facial expression classification.

### 3.1. Sparse Mask Attention Learning

According to facial action units, some regions of the face, such as the eyebrows and mouth, are more important for determining the expression category. Therefore, SMAL is used to obtain more comprehensive and richer key features, so that the interference of useless information can be eliminated to enhance the feature extraction ability of our network. This module is composed of sparse mask learning and channel attention. Sparse mask learning can learn sparse face regions and dynamically apply convolution to achieve spatial adaptation, which increases few parameters but improves performance. The introduced channel attention can enhance the cross-dimension features and the global features. Thus, SMAL can extract more effective spatio-channel interaction information. Furthermore, the SMAL module is connected to residual functions [15] to solve the problem of gradient disappearance during network training without adding extra parameters. The specific process is shown in Figure 2.

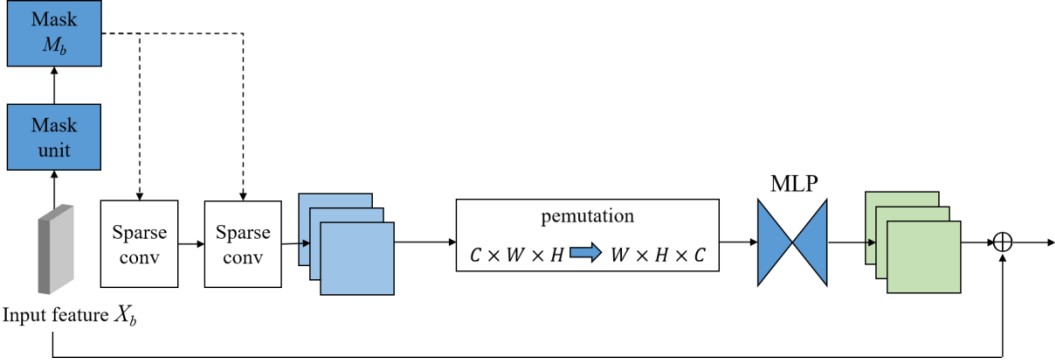

**Figure 2.** The structure of Sparse Mask Attention Learning.

The input of SMAL is the features processed by a $3 \times 3$ convolution and max-pooling with the stride of 2. Then, the mask is obtained through the mask unit according to the input feature, and sparse convolution is performed on the mask to gain sparse features. Next, the obtained sparse features are sent to the channel attention sub-module, which uses three-dimensional permutation to retain information across three dimensions. It magnifies cross-dimension channel dependencies with a two-layer Multi-Layer Perceptron (MLP) to output more effective channel information of facial expression. Furthermore, SMAL is connected with residual functions, and the operation of the residual block is described by Equation (1). The residual block consists of 2 convolutions with the BN and ReLU functions sequentially. The residual learning result is added to the mapped result, and then the ReLU activation function is used to obtain the final output.

$$X_{b+1} = \sigma(X_b + F(X_b)) \tag{1}$$

where $X_b$ represents the input, $F(\cdot)$ denotes the residual mapping function, and $\sigma$ is the activation function. In addition, mask $M_b$ of the input image is generated according to the input $X_b$ through Mask Unit $M$, for defining the spatial position where the convolution operation should be performed. Mask Unit $M$ performs the squeeze operation and the $3 \times 3$ convolution on the spatial dimension to generate the corresponding mask from the input image. Through a global Ave-Pooling and FC layer, the squeeze operation creates the weight matrix with a $1 \times 1$ spatial dimension over all channels. Moreover, the weight matrix extracted by the squeeze operation and the $3 \times 3$ convolution is added to obtain the corresponding masks $M_b \in R^{w_{b+1} \times h_{b+1}}$.

Since the pixel values in the mask $M_b$ generated by the Mask Unit $M$ cannot be optimized by backpropagation, the Gumbel-Softmax trick [50] is used for each pixel of mask $M_b$ to obtain a continuous differentiable mask $G_b$ in an end-to-end manner, as shown in Equation (2). After performing residual learning on mask $G_b$, the calculation of the sparse mask sub-module is as shown in Equation (3).

$$G_b = \varsigma(M(X_b)) \tag{2}$$

$$X_{b+1} = \sigma(X_b + F(X_b) \circ G_b) \tag{3}$$

where $\varsigma(\cdot)$ indicates the Gumbel-Softmax trick and $\circ$ represents element-wise multiplication in the spatial dimension $(w_{b+1} \times h_{b+1})$. In order to change the pixel values from $m \in (-\infty, +\infty)$ to $(0, 1)$, the calculation is shown in Equation (4).

$$y_i = \frac{\exp((\log(\rho_i) + g_i)/\tau)}{\sum_{j=1}^{k} \exp((\log(\rho_j) + g_j)/\tau)} \tag{4}$$

where $g_i$ is the noise sample extracted from the Gumbel distribution to enhance the anti-interference ability of the network training. Each pixel of the generated mask represents

the probability $\rho_i$ of performing the convolution operation, as shown in Equation (5). On the contrary, the probability $\rho_i$ of not performing the convolution operation is as shown in Equation (6). They are carried into Equation (4) to obtain the result of the binary sample as $k = 2, i = 1$, which is shown in Equation (7).

$$\rho_1 = \sigma(m) \tag{5}$$

$$\rho_2 = 1 - \sigma(m) \tag{6}$$

$$y_1 = \sigma(\frac{m + g_1 + g_2}{\tau}) \tag{7}$$

where $\tau$ denotes the fixed hyper-parameter with size as 1. Therefore, the final calculation of discrete samples $z$ for forward propagation (upper) and back propagation (lower) is shown in Equation (8).

$$z = \begin{cases} y_1 > 0.5 \equiv \frac{m+g_1+g_2}{\tau} > 0 \\ y_1 \end{cases} \tag{8}$$

In order to extract more effective information, we introduce channel attention after sparse mask learning. The previous channel attention network ignored the importance of cross-dimensional interaction, while the channel attention that we adopted can capture important features in three dimensions. The information of three dimensions is first arranged and combined by 3D permutation. Then, a two-layer MLP with an encoder–decoder structure is applied to enhance the channel dependence, which is composed of two FC layers and ReLU activation function. The calculation of channel attention is shown in Equation (9).

$$F_{out} = M_c(F_{in}) \otimes F_{in} \tag{9}$$

where $F_{in}$ represents the feature map obtained by sparse mask learning and $M_c(\cdot)$ indicates the operation of channel attention. The weights obtained by channel attention are multiplied with the feature matrix of the sparse mask to obtain the final output.

Finally, the sparse loss [25] is applied to conduct joint training with the cross-entropy loss of the whole network for learning the effective sparse mask during the training process. By setting a computational budget $\theta \in [0, 1]$ to define the relative amount of desired convolution operations performed on the generated mask, the process of calculating sparse loss is shown in Equations (10)–(14). Among them, Equation (10) is the final loss function. Equation (11) minimizes the difference between the given computational budget $\theta$ and the budget used by a network consisting of $B$ residual blocks. The ratio of the number of convolution operations between the SMAL module and the original mask learning module is shown in Equation (12). In order to obtain better initialization values, there is an upper bound and a lower bound in the sparse loss function. Equations (13) and (14) are the definitions of the upper and lower bounds, respectively.

$$L = L_{classify} + \alpha(L_{sp,net} + L_{sp,lower} + L_{sp,upper}) \tag{10}$$

$$L_{sp,net} = (\frac{\sum_b^B F_{b,sp}}{\sum_b^B F_b} - \theta)^2 \tag{11}$$

$$\rho_b = F_{b,sp}/F_b \tag{12}$$

$$L_{sp,lower} = \frac{1}{B}\sum_b^B \max(0, p \cdot \theta - \rho_b)^2 \tag{13}$$

$$L_{sp,upper} = \frac{1}{B}\sum_b^B \max(0, \rho_b - (1 - p(1 - \theta)))^2 \tag{14}$$

where $L_{classify}$ is the cross-entropy loss function of this paper, $\alpha(L_{sp,net} + L_{sp,lower} + L_{sp,upper})$ is the sparse loss function, $F_{b,sp}$ denotes the number of floating point operations for con-

volutions in the SMAL module, $F_b$ indicates the number of floating point operations for convolutions in the original mask learning module, $\rho_b$ is the ratio between $F_{b,sp}$ and $F_b$, and $p$ is a fixed cosine annealing hyper-parameter. In this paper, $B$ is set to 1 and $p$ is set to 0.33. When $\theta = 0.7$, this module can achieve good results based on actual experiments.

### 3.2. Multi-Scale Attention

Inspired by the literature [26], MSA is utilized to extract richer global features of facial expressions. This module can extract more abundant multi-scale features at a more fine-grained level. The structure of MSA is shown in Figure 3.

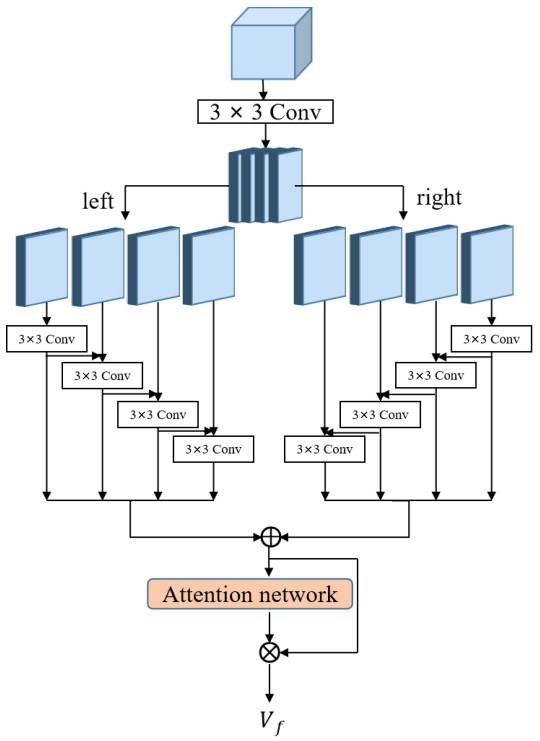

**Figure 3.** The structure of Multi-Scale Attention module.

The feature maps obtained by the previous processing are put into a $3 \times 3$ convolution and max-pooling to obtain the feature map $X$. Then $X$ will be divided along the channel dimension into $n$ subsets of the feature map, that is, $X_i, i \in \{1, 2, \cdots, n\}$. Each subset $X_i$ has $\frac{1}{n}$ channel and its spatial size is the same as the original feature map $X$. Next, each $X_i$ is processed with a $3 \times 3$ convolution $P_i^l(\cdot)$ operation from the left and right directions to receive more abundant multi-scale features. A symmetrical structure is introduced to learn multi-scale features, which can ensure that the feature subsets at the front and back both can contain richer scale information. The attention network is the same as that in the SMAL module. The specific procedure of MSA is shown in Equations (15) and (16). The subsets $X_i$ are sorted from left to right as $1, 2, \cdots, n$.

$$Y_i^{left} = \begin{cases} p_i^{left}(X_i) & i = n \\ P_i^{left}(X_i + Y_{i-1}^{right}) & 1 < i \leq n \end{cases} \tag{15}$$

$$Y_i^{right} = \begin{cases} p_i^{right}(X_i) & i = n \\ P_i^{right}(X_i + Y_{i+1}^{right}) & 1 \leq i < n \end{cases} \tag{16}$$

where $P_i^l(\cdot)$ represents the $3 \times 3$ convolution operation, $l \in \{left, right\}$ of $P_i^l(\cdot)$ represents the location, $Y_i^{left}$ indicates the output of the left, and $Y_i^{right}$ indicates the output of the right. From Equation (15), we can notice that each operation of $p_i^{left}(\cdot)$ can capture features from

all subsets $\{X_j, j \leq i\}$. From Equation (16), we can notice that each operation of $p_i^{right}(\cdot)$ can capture features from all subsets $\{X_j, n \geq j \geq i\}$. Each $Y_i^l$ contains a different number and scale of subset features. In order to obtain diversified multi-scale features, all $Y_i^l$ are combined along the channel dimension. The calculation of final output $Y_i$ is shown in Equation (17).

$$Y_i = Y_i^{left} + Y_i^{right} \qquad (17)$$

Considering the number of parameters, the number of subsets in this paper is set to 4. After obtaining the final output, channel attention is introduced to further enhance the interaction of cross-dimensional features for making the network focus on the more discriminative and significant regions of facial expression images.

### 3.3. Relational Attention

The scene coding stream is mainly used to extract the contextual cue features except for facial regions. The context coding module adopts the same backbone network as used in the literature [24], which is composed of a 5-layer $3 \times 3$ convolution with BN and ReLU and 4-layer max-pooling in a stride of 2. In order to construct the relationship between the face and the context, the attention mechanism is utilized in the RA module to capture the similarity between them for building the hidden relationship [28]. Specifically, the facial expression features extracted from the face coding stream and the context features extracted from the scene coding module are taken as inputs of RA. Then, the final output of the scene coding stream is obtained by multiplying the normalized attention map with the extracted context features. The attention map represents the priority of each position in the feature map. The structure of the RA module is shown in Figure 4.

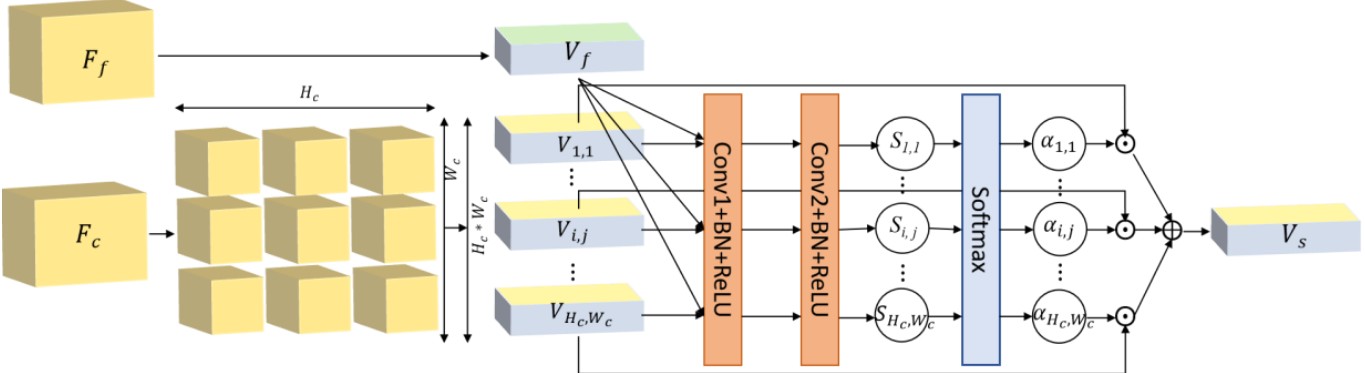

**Figure 4.** The structure of Relational Attention module.

First, the feature maps $F_f$ of faces extracted from the face coding stream are transformed into the feature vectors $V_f$ by the global ave-pooling operation. The contextual feature map $F_c \in R^{H_c \times W_c \times D_c}$ is a 3D tensor, while $H_c$, $W_c$, and $D_c$ represent the height, width, and channel dimension of the feature map, respectively. $F_c$ is divided into a set of $H_c \times W_c$ vectors with $D_c$ dimensions, and each vector in each cell $(i, j)$ represents the embedded features at that location. At each location $(i, j)$ of $F_c$, we have vectors $F_c^{(i,j)} = v_{i,j}, v_{i,j} \in R^{D_c}$ and $1 \leq i \leq H_c, 1 \leq j \leq W_c$. The representation of $F_c$ is shown in Equation (18).

$$F_c = \left\{ v_{i,j} \in R^{D_c} \,\middle|\, 1 \leq i \leq H_c, 1 \leq j \leq W_c \right\} \qquad (18)$$

where $v_{i,j}$ represents the feature vector of contextual feature map $F_c$ at the $(i, j)_{th}$ position. A feature matrix containing both the features of the face and context is obtained by concatenating $V_f$ and $v_{i,j}$ along the channel dimension. Then the concatenated vectors are input into a two-layer $3 \times 3$ convolution to calculate the raw score values $s_{i,j}$ related to the region. The Softmax function is then used to generate the attention graphs, as shown in

Equation (19). Finally, the attention maps $\alpha_{i,j}$ are weighted into the corresponding $v_{i,j}$ to acquire the final representation vectors $V_s$, as shown in Equation (20).

$$\alpha_{i,j} = \frac{\exp(s_{i,j})}{\sum_a \sum_b \exp(S_{a,b})} \tag{19}$$

$$V_s = \sum_i \sum_j (\alpha_{i,j} \odot v_{i,j}) \tag{20}$$

where $\odot$ represents the element-wise multiplication operation. Through the above process, the RA module can guide the network to pay attention to those $v_{i,j}$ that represent more important and discriminative regions. As a result, the other irrelevant areas in the image are ignored.

### *3.4. Adaptive Fusion Module*

In order to carry out the final classification, it is necessary to fuse facial features and context features. Therefore, an adaptive fusion module is designed, as shown in Figure 1. Our fusion model can adaptively learn and infer an optimal fusion weight according to extracted multi-cues features. The learned weights are applied to the corresponding features for concatenating, fusion, and classification.

The facial features vector $V_f$ and contextual features vector $V_s$ obtained by the two-stream network are used as the inputs of the adaptive fusion network. The two feature vectors are first input into two FC layers and the Softmax layer to obtain the facial feature weight $\lambda_F$ and the contextual feature weight $\lambda_s$. Then, the learned weights are normalized by a Softmax function. Next, $\lambda_F$ and $\lambda_s$ are respectively applied to corresponding feature vectors to enhance important features and suppress redundant features. At last, the two parts of features are concatenated and input into the two FC layers followed by a Softmax layer to obtain the final recognition results, as shown in Equation (21).

$$X_A = \Pi(V_f \odot \lambda_F, V_s \odot \lambda_s) \tag{21}$$

where $\Pi$ represents the concatenate operation.

## 4. Results

In this section, we conduct a series of experiments to validate the effectiveness of our MCF-Net for facial expression recognition. Furthermore, the proposed method is compared with other FER methods on two standard datasets, CAER-S and NCAERS.

### *4.1. Experimental Dataset*
#### 4.1.1. CAER-S Dataset

The CAER-S dataset [24] is a static dataset of large-scale natural scenes, which contains human faces and background information. The dataset was collected from 79 TV shows and each video clip was manually annotated with six emotion categories, including "anger", "disgust", "fear", "happy", "sad", "surprise", as well as "neutral". In total, 13,201 clips and approximately 1.1M frames were available. Then, approximately 70K static images are extracted to create a static image subset, wherein the size of each image is $712 \times 400$. The dataset is randomly split into training (70%), validation (10%), and testing (20%) sets. Sample frames of the CAER-S dataset are illustrated in Figure 5. The number of training, validation, and testing sets in this paper are similar to the CAER-S database.

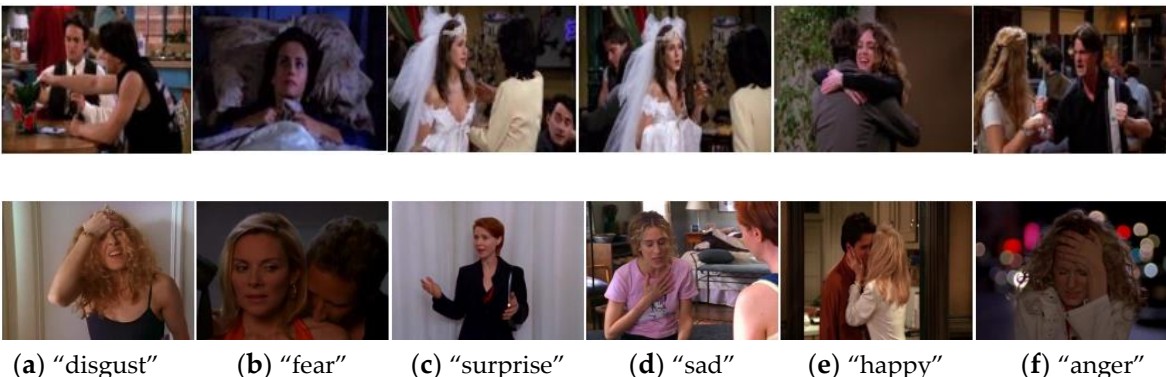

| (a) "disgust" | (b) "fear" | (c) "surprise" | (d) "sad" | (e) "happy" | (f) "anger" |

**Figure 5.** Sample frames of the CAER-S dataset.

4.1.2. NCAER-S Dataset

In CAER-S dataset, many images in the training and the testing sets are extracted from the same video, which makes them very similar to each other. In order to improve the robustness of the model, the NCAER-S dataset [28] was extracted from the CAER video clips to deal with the above issue. Each video in the original CAER dataset is split into multiple parts, and each part is approximately 2 s long. This dataset assures that the training frames and testing frames are never from one original input video. The statistics of the original CAER and the NCAER-S training sets are shown in Figure 6.

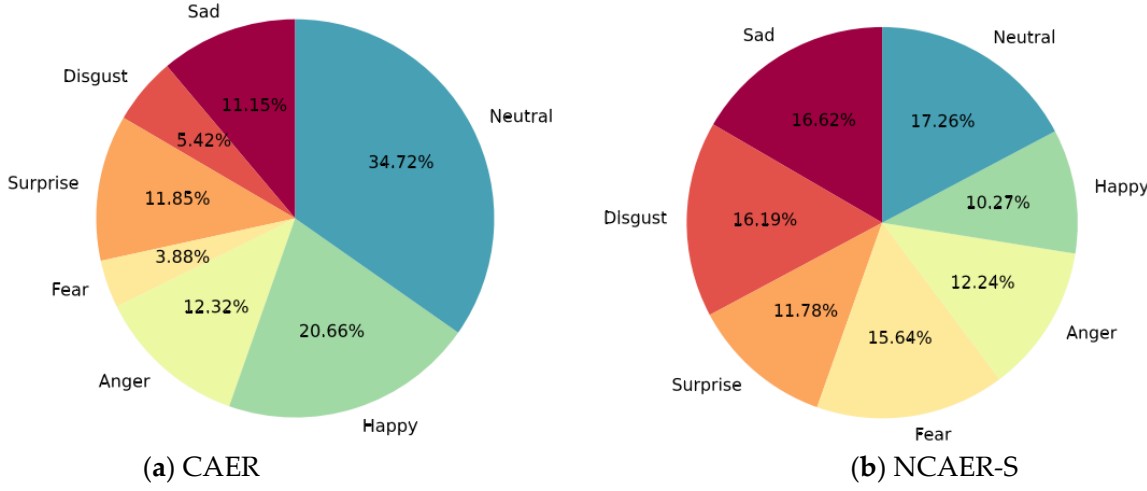

| (a) CAER | (b) NCAER-S |

**Figure 6.** Percentage of each emotion category in CAER and NCAER-S training sets.

*4.2. Experiments and Result Analysis*

4.2.1. Experiments and Result Analysis of MCF-Net on CAER-S Dataset

In order to verify the effectiveness of MCF-Net, our model is tested and validated on the CAER-S dataset. The accuracy curve of MCF-Net is shown in Figure 7, in which the red line indicates the training accuracy, and the blue line represents the validation accuracy. It can be seen from Figure 7 that both training accuracy and verification accuracy of MCF-Net gradually improved. The accuracy begins to converge around the epoch of 30 and tends to be stable around the epoch of 45, which shows that the proposed MCF-Net is reasonable and effective.

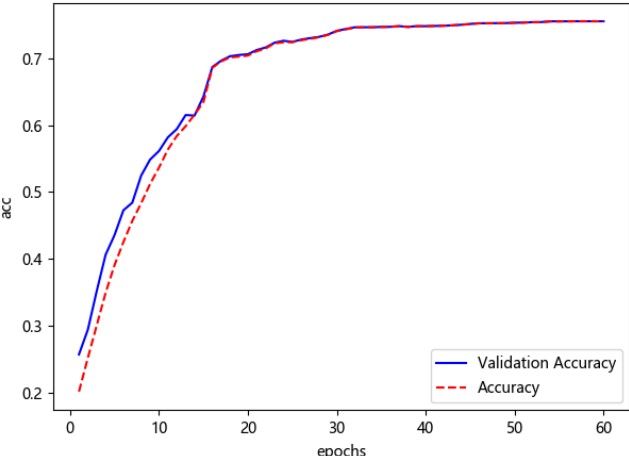

**Figure 7.** The accuracy curve of MCF-Net on CAER-S dataset.

　　　The loss function curve of the MCF-Net obtained during the training process is shown in Figure 8, where the red line represents the training loss, and the blue line means the validation loss. It can be concluded that both training loss and validation loss of our model gradually decrease and begin to converge around the epoch of 30. At the epoch of 45, the curve tends to be stable, and the loss value no longer changes, finally stabilizing at 0.71. The results prove that MCF-Net can better fit the distribution of samples in the training set and the validation set, which verify the rationality of the joint loss at the same time.

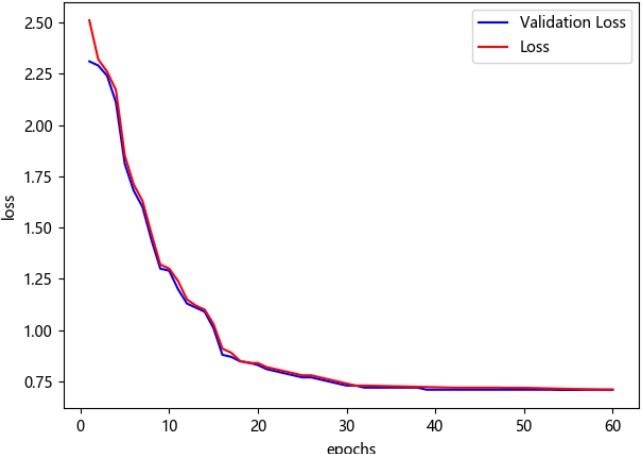

**Figure 8.** The loss curve of MCF-Net on CAER-S dataset.

　　　Our model is compared with some classical and SOTA methods on the CAER-S dataset, as shown in Table 1. It can be seen that the accuracy of MCF-Net is higher than that of baseline CAER-Net-S by 2.17% and much higher than those of classical methods, with only a small increase in parameters. In order to improve the performance of recognition, ResNet-18 is adopted to replace the shallow convolution and pooling layers of the face coding stream. When compared with other recent state-of-the-art methods, we obtained state-of-the-art performance on the CAER-S dataset with 81.82% classification accuracy. Although the parameters of MCF-Net(ResNet18) have been increased, it achieves a great improvement in recognition accuracy. These results confirm that adding three modules can effectively encode both facial information and context information to improve the facial expression classification results. Furthermore, the parameters prove that the training time consumption of our model is short, and the computational cost is less. To specify, the calculation cost is reported in Multiply-Accumulates (MAC), averaged over all test images. From Table 1, it can be seen that the calculation costs of our MCF-Net and the

original CAER-Net-S are 369MMAC and 385MMAC, respectively. Besides, the calculation cost decreases as the accuracy of our model improved.

**Table 1.** Comparison results on CAER-S dataset.

| Methods | Params (M) | MAC (M/G) | Accuracy (%) |
|---|---|---|---|
| ImageNet-AlexNet [13] | ~89.92 | 102G | 47.36 |
| ImageNet-VGGNet [14] | ~102.34 | 198G | 49.89 |
| ImageNet-ResNet [15] | ~132.54 | 247G | 57.33 |
| Fine-tuned AlexNet [13] | ~89.92 | 102G | 61.73 |
| Fine-tuned VGGNet [14] | ~102.34 | 198G | 64.85 |
| Fine-tuned ResNet [15] | ~132.54 | 247G | 68.46 |
| CAER-Net-S [24] | ~2.39 | 385M | 73.51 |
| Kosti et al. [51] | ~27.86 | 78G | 74.48 |
| Gao et al. [52] | – | – | 81.31 |
| Zeng et al. [53] | ~68.24 | 115G | 81.31 |
| Zhao et al. [45] | – | – | 81.48 |
| MCF-Net | ~4.25 | 369M | 75.68 |
| MCF-Net(ResNet18) | ~14.2 | 756M | 81.82 |

In order to prove the recognition effect of the improved model on each category of samples, the confusion matrix of CAER-Net-S and MCF-Net (ResNet18) on CAER-S datasets is shown in Figures 9 and 10, respectively. In the confusion matrix, the horizontal represents the predicted label while the vertical indicates the real label. The values of the diagonal show the classification accuracy of the network for each category, and darker colors represent higher accuracy.

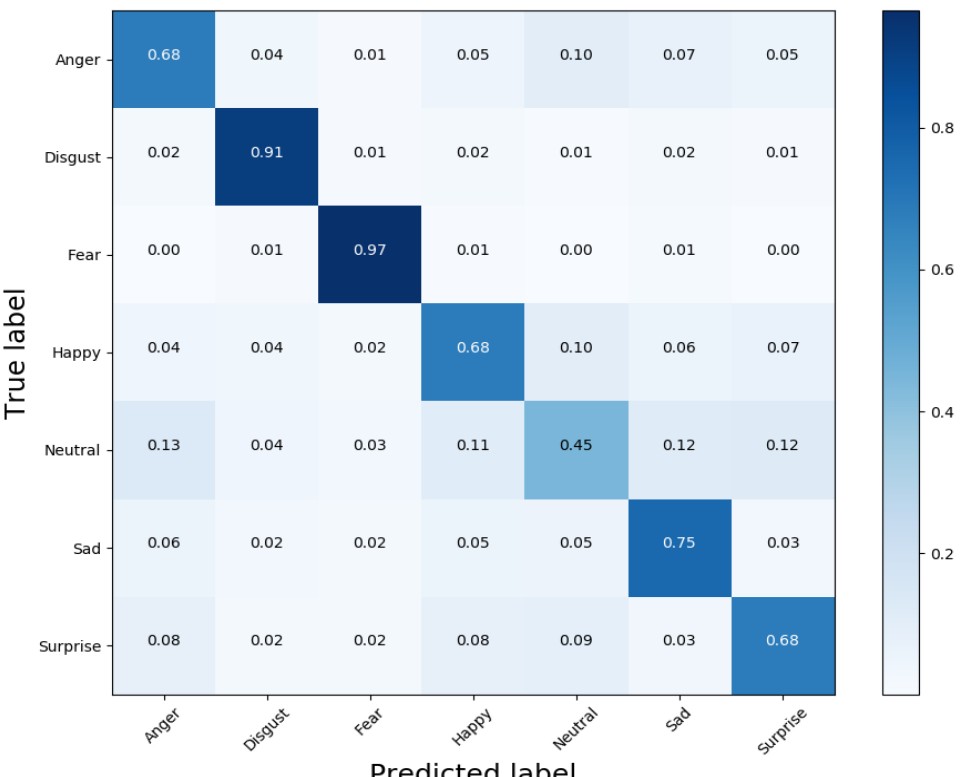

**Figure 9.** The confusion matrix of CAER-Net-S on CAER-S dataset.

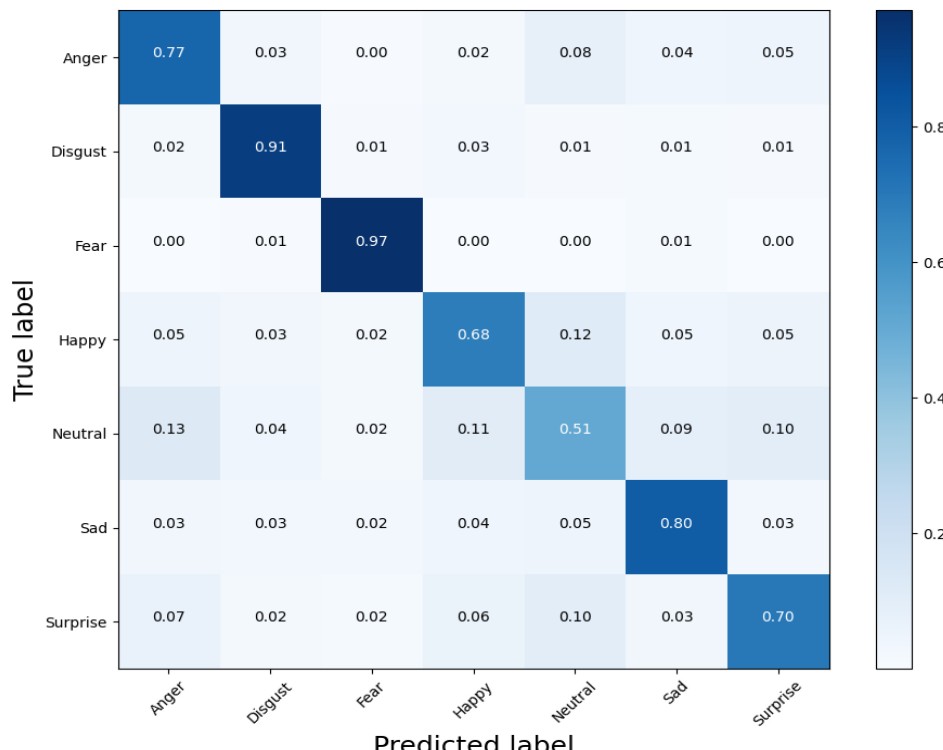

**Figure 10.** The confusion matrix of MCF-Net(ResNet18) on CAER-S dataset.

It can be seen from Figures 9 and 10 that MCF-Net(ResNet18) has the highest accuracy for the fear category, reaching 0.97, and the lowest accuracy for the neutral category, with only 0.51. It is because neutral expressions are easily confused with other categories of expressions. Compared with the confusion matrix of CAER-Net-S, our model significantly improved the classification accuracy of each category. Among them, the accuracy of "anger", "neutral", "sad", and "surprise" increased by 0.09, 0.06, 0.05, and 0.02, respectively. Therefore, the confusion matrix further tests the validity of MCF-Net(ResNet18).

### 4.2.2. Ablation Experiment and Result Analysis of MCF-Net on CAER-S Dataset

Ablation experiments are carried out on the CAER-S dataset. In this experiment, the effectiveness of the sparse mask attention learning module, multi-scale attention module, relational attention module, and channel attention in our network architecture are proven, respectively. The comparison results are shown in Table 2, where the baseline method is the two-stream network of the 5-layer $3 \times 3$ convolution with BN and ReLU and 4-layer max-pooling in the stride of 2. Compared with the baseline method, the three modules have 0.7%, 1.3%, and 2.17% improvement, respectively. In addition, the effectiveness of channel attention is also checked. SMAL is higher than that of SML by 0.24%, and MSA is higher than that of MS by 0.09%. The results of ablation experiments identify the availability of each module for improving the performance of facial expression recognition.

**Table 2.** The comparison results of ablation experiments on the CAER-S dataset.

| Methods | Params (M) | Accuracy (%) |
| --- | --- | --- |
| Baseline | ~2.39 | 73.51 |
| Baseline+SML | ~2.44 | 73.97 |
| Baseline+SMAL | ~2.45 | 74.21 |
| Baseline+SMAL+MS | ~3.92 | 74.72 |
| Baseline+SMAL+MSA | ~3.96 | 74.81 |
| Baseline+SMAL+MS+RA | ~4.22 | 75.21 |
| Baseline+SMAL+MSA+RA | ~4.25 | 75.68 |

According to the key feature information of the context captured in the scene coding stream, visual attention heat maps can be obtained, as shown in Figure 11. It can be seen that our model can capture more significant contextual regions related to facial expression recognition, such as the face of other people and hands et al., which further testifies to the effectiveness of the RA module. At the same time, it also proves that the performance of the model can be improved effectively by utilizing the multi-cue feature information.

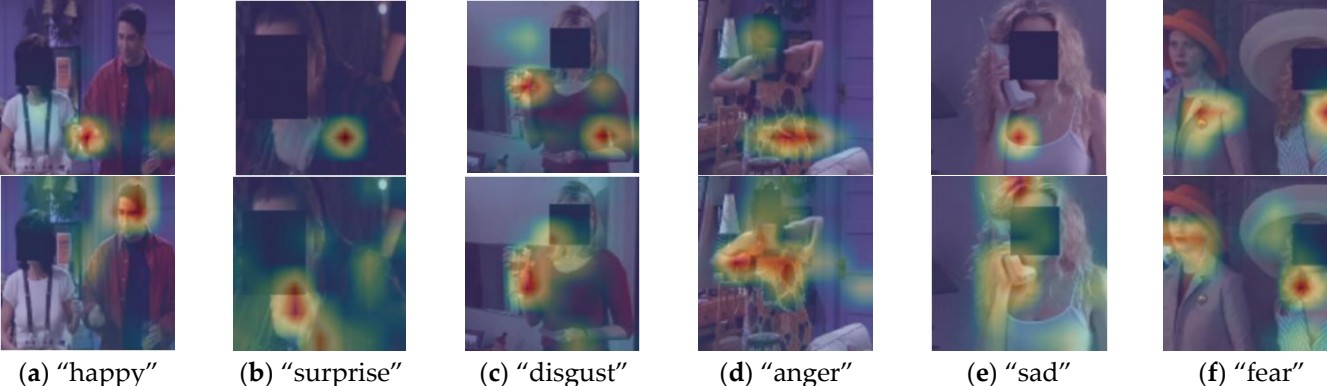

(**a**) "happy"  (**b**) "surprise"  (**c**) "disgust"  (**d**) "anger"  (**e**) "sad"  (**f**) "fear"

**Figure 11.** Visual attention heat maps on the CAER-S dataset. The first row shows the heat maps of CAER-Net-S. The second row illustrates the heat maps of our model.

### 4.2.3. Multi-Cue Feature Weights of Adaptive Fusion Network

The face weights $\lambda_F$ and context weights $\lambda_C$ of some samples learned by the adaptive fusion network on the CAER-S dataset are shown in Figure 12. It can be seen that our model can adaptively learn more important cues according to the input image. In Figure 11b,f, most of the face regions have been occluded, which provide less feature information. Besides, the feature information in the context, such as the gesture or interaction with others, has a great impact on expression recognition. Therefore, in view of this situation, our fusion network makes the context weights greater. The learned weights $\lambda_F$ and $\lambda_C$ are multiplied with the face features and context features extracted from the two-stream network, so as to strengthen the corresponding key features and suppress redundant features.

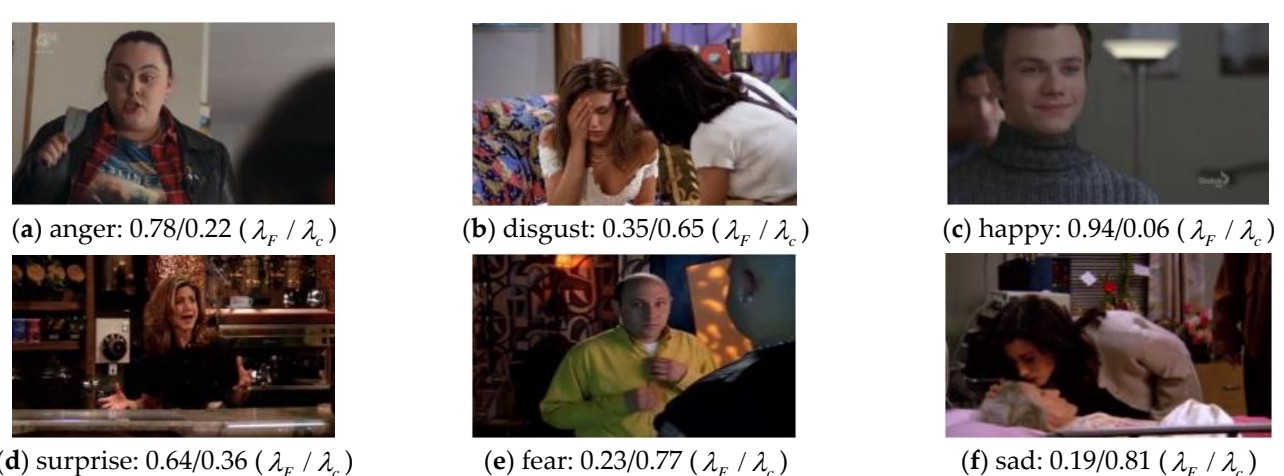

(**a**) anger: 0.78/0.22 ($\lambda_F$ / $\lambda_c$)  (**b**) disgust: 0.35/0.65 ($\lambda_F$ / $\lambda_c$)  (**c**) happy: 0.94/0.06 ($\lambda_F$ / $\lambda_c$)

(**d**) surprise: 0.64/0.36 ($\lambda_F$ / $\lambda_c$)  (**e**) fear: 0.23/0.77 ($\lambda_F$ / $\lambda_c$)  (**f**) sad: 0.19/0.81 ($\lambda_F$ / $\lambda_c$)

**Figure 12.** The weights of some samples on the CAER-S dataset.

### 4.2.4. Experiments and Result Analysis of MCF-Net on NCAERS Dataset

The NCAERS dataset belongs to a subset of the CAER-S dataset. Since some images of the training set and the testing set in the CAER-S dataset are extracted from the same video sequence, there is a certain data similarity, which may reduce the robustness of the model. In order to further verify the significance and robustness of MCF-Net, comparative

experiments are conducted with other methods on the NCAERS dataset. The experimental results are shown in Table 3. From Table 3, it can be seen that our model is superior to other classical methods and the baseline method, which shows the effectiveness and robustness of the proposed MCF-Net.

**Table 3.** Comparison results using NCAER-S dataset.

| Methods | Accuracy (%) |
| --- | --- |
| VGG16 [14] | 42.85 |
| ResNet50 [15] | 41.41 |
| CAER-Net-S [24] | 44.14 |
| MCF-Net | 45.59 |

## 5. Conclusions

In this work, we presented a novel model to exploit FER more efficiently by using the proposed Multi-cues Fusion Net (MCF-Net) with the two-stream structure. Experiments have shown that our model can improve the accuracy of facial expression recognition in the wild compared with the current state-of-the-art results. We report the results of our MCF-Net with two different backbones: The original encoding structure and ResNet-18 of the face coding stream. A large number of experimental results on the CAER-S and NCAER-S datasets consistently demonstrate the effectiveness and robustness of our model. However, illumination and contrast can vary in different images even from the same person with the same expression, especially in unconstrained environments, which can result in large intra-class variances. In future work, we will try to solve the problems related to large intra-class variances and blurred images by means of intra-class and inter-class similarity processing, so as to further enhance the robustness of the model and improve the generalization ability of the model. In addition, we will try more SOTA backbones to further improve the performance of our model, such as a transformer. The recognition accuracy of replacing Resnet-18 with other backbones will be tested.

**Author Contributions:** Data curation, X.K.; formal analysis, H.X.; funding acquisition, J.K. and J.L.; investigation, J.L.; methodology, H.X.; resources, J.K.; writing—original draft, H.X. and J.W.; writing—review and editing, J.W.; conceptualization, H.X. All authors have read and agreed to the published version of the manuscript.

**Funding:** This research was supported by the National Natural Science Foundation of China (62272096) and the Fund of Jilin Provincial Science and Technology Department (20210201077GX).

**Institutional Review Board Statement:** Not applicable.

**Informed Consent Statement:** Not applicable.

**Data Availability Statement:** Not applicable.

**Conflicts of Interest:** The authors declare no conflict of interest.

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
