# Peer review of "MCF-Net: Fusion Network of Facial and Scene Features for Expression Recognition in the Wild"

_applsci, doi:10.3390/app122010251_

Round 1

Reviewer 1 Report

Put actual number of performance in Abstract, for example we are getting 95% accuracy.

3.1 is started after section 2, How it is possible??

Keep formatting of paper consistence throughout the paper.

Add more literature review from 2022 papers.

Author Response

Thanks for your constructive comments and suggestions on our manuscript.

  1. We have put actual number of performance in Abstract. Please see the contents marked with red color on Page 1 in our revised paper.
  2. We have revised the chapter number. Please see the contents marked with red color on Page 3-4 in our revised paper.
  3. We have thoroughlychecked the format of our paper.
  4. We have added the following referencesabout expression recognition and added the corresponding brief description for each reference, please see the contents marked with red color on Pages 4 in our revised paper. For the sake of convenience, we attached these new additions below:

[35] proposed an occluded expression recognition model based on the generated countermeasure network, with two modules, namely, occluded face image restoration and face recognition. [48] adopted Facial Landmark Detection (FLD) as the auxiliary task and explored new multi-task learning strategies for FER. [49] proposed an Adaptive Correlation Loss to guide the network towards generating embedded feature vectors with high correlation for within-class samples and less correlation for between-class samples.

Reference:

[35] Ge, H.; Zhu, Z.; Dai, Y. et al. Facial expression recognition based on deep learning. Computer Methods and Programs in Biomedicine. 2022, 215, 106621.

[48] Yu, W.; Xu, H. Co-attentive multi-task convolutional neural network for facial expression recognition. Pattern Recognition: The Journal of the Pattern Recognition Society. 2022, 123, 108401.

[49] Fard, A. P.; Mahoor, M. H. Ad-Corre: Adaptive Correlation-Based Loss for Facial Expression Recognition in the Wild. IEEE Access. 2022, 10, 26756-26768.

Reviewer 2 Report

a. Propose Multi-cues Fusion Net (MCF-Net), a novel deep learning model with two-stream structure for FER. With adequate results. Interesting and appropriate work for the journal.

b. It is considered an original topic due to the way in which the problem is approached: proposes a two- stream coding network to extract face and scene representations. Then, an adaptive fusion module 20 is employed to fuse the two different representations for final recognition. There is a good background check and similar jobs.

c. In general, the text is clear and easy to read. It is recommended to review the wording to improve it. The research design is appropriate and an adequate development of the project is presented.

d. Through the results and their discussion, it can be observed that there is verification of the objective and research questions presented in the article. The results presented are interesting and there is a correct discussion of them.

e. It is recommended to make the contribution clear in the title.

f. Justify the method of experimentation. It presents an adequate method of experimentation supported by similar works.

Author Response

  1. Thanks for your positive comments.
  2. Thanks for your positive comments.
  3. We have carefully revisedthe English writing of our paper, please see the contents marked with red color.
  4. Thanks for your positive comments.
  5. Wehave changed the title to “MCF-Net: Fusion Network of Facial and Scene Features for Expression Recognition in the Wild”.
  6. Thanks for your positive comments.

Reviewer 3 Report

This manuscript presents a novel deep-learning-based FER method termed MCF-Net. The method comprises two streams for coding facial features and scene features. Several novel attention modules are employed in both streams. The experimentally obtained results using two public FER benchmark datasets revealed that the proposed method outperformed the 11 previous methods. Nevertheless, the questions below remain subjects that must be addressed in this manuscript.

1. ResNet-18 is used in the Face Coding Stream. Is this backbone an appropriate choice among the numerous types of ConvNets? This is because ResNet is a slightly older backbone. Have the authors considered recent high performance backbones such as InceptionResNetV2, ResNeXt, DenseNet, MobileNet, NASNet, EfficientNetV2, RegNet, or ConvNeXt?

2. Facial expressions are composed of time-series features. How does the proposed method handle time-series features?

3. Recently, transformer backbones have outperformed ConvNet backbones in various applications. Is your method superior to the following method based on transformers?

Context-LGM: Leveraging Object-Context Relation for Context-Aware Object Recognition Mingzhou Liu, Xinwei Sun, Fandong Zhang, Yizhou Yu, Yizhou Wang, arXiv:2110.04042, https://doi.org/10.48550/arXiv.2110.04042

4. In line 128, the fourth contribution claims "A large number of experiments". However, the experimental results using the NCAERS dataset are limited to seven lines of text and Table 3 in Section 4.2.4. Why didn't you provide the similarly detailed results as those of the CAER-S dataset?

5. The network size is shown in Table 1. However, there is no information about the calculation cost. Real-time processing is one of the key factors for FER applications in the wild.

6. Check the first word of line 245.

7. The abbreviation used in line 438 would be inappropriate in an academic paper.

Author Response

Thanks for your constructive comments and suggestions on our manuscript.

  1. The innovation of this paper is thetwo-stream structure for extracting both facial and scene information, which comprises Multi-scale Attention module and Relational Attention module. Therefore, even if the slightly older Resnet-18 is used as the backbone of the Face Coding Stream, our recognition results are superior to other algorithms. 

    During the experiments, we tested the effect of different Encoding network architectures. Specifically, the MobileNetV2 and ResNet-18 are adopted as the backbone network to extract features in the Face Coding Stream, respectively. According to the experimental results, we observe that the ResNet-18 significantly outperforms other shallower architectures (Original and MobileNetV2). And the backbone network of literature [46] is EfficientNet, it can be seen from Table 1 that ResNet-18 is a suitable backbone for extracting facial features.

    In addition, we will try more sota backbones to further improve the performance of our model. In future work, the recognition accuracy of replacing Resnet-18 with other backbones will be tested. We have added related contents in the section “Conclusions” in the revised manuscript. Please see the contents marked with red color in the section “Conclusions” in the revised manuscript.

  1. The proposed model is applied to the static images of large-scale natural scenes, so it is unnecessary to consider the time features. If our model is extended to videos,  each frame in the video can be regarded as a static image to recognize the facial
  2. In the paper you recommended, transformer has achieved good results in object recognition. Although transformer can solve the problem that the local receptive field of CNN is limited, it can effectively obtain the global information and enhance the  expression ability of models. However, the key features in facial expression recognition are often concentrated facial regions and other critical scene regions. Hence, we use Multi-scale Attention module to extract multi-scale global features at a fine-grained level.

     As you mentioned, real-time processing is one of the key factors for FER applications in the wild. At present, one of the important factors that transformer cannot replace CNN is the computational efficiency. The recognition methods in the field of computer vision generally use the transformer structure in NLP, but the amount of information in images is far greater than that in text, so the calculation cost of transformer is still large. Consequently, there is no paper using transformer for facial expression recognition. In the future work, we will consider combining transformer with the proposed model to further improve the recognition accuracy. We have added related contents in the section “Conclusions” in the revised manuscript. Please see the contents marked with red color in the section “Conclusions” in the revised manuscript.

  1. NCAER-S dataset is a new dataset collected and established in 2021. At present, there are few comparative experimental results on this dataset. In order to improve the robustness of our model, comparative experiments are conducted with several exiting methods on the NCAERS dataset.
  2. The parameters prove that the training time consumption of our model is short and the computational cost is less. To specify, our calculation cost is reported in Multiply-Accumulates(MAC), averaged over all test images. The calculation costs of our MCF-Net and original CAER-Net-S are 369MMAC and 385MMAC, respectively. It can be seen that the calculation cost is decreasing while the accuracy of our model is improved. We have added the calculation cost to the revised paper, please see Table 1, and the contentmarked with red color on Page 13.
  3. We have checked and revised the first word of line 245, please see the content marked with red color on Page 6.
  4. We have revised the abbreviation used in line 438, and corresponding abbreviations in Figures 1 and 4 have also been changed. Please see the contentmarked with red color on Page 11.
